# Availability of mRNA Obtained from Peripheral Blood Mononuclear Cells for Testing Mutation Consequences in Dystrophic Epidermolysis Bullosa

**DOI:** 10.3390/ijms222413369

**Published:** 2021-12-13

**Authors:** Eijiro Akasaka, Hajime Nakano, Daisuke Sawamura

**Affiliations:** Department of Dermatology, Hirosaki University Graduate School of Medicine, 5 Zaifu-cho, Hirosaki 036-8562, Japan; akasakae@hirosaki-u.ac.jp (E.A.); Smartdai@hirosaki-u.ac.jp (D.S.)

**Keywords:** dystrophic epidermolysis bullosa, type VII collagen: mRNA, mesenchymal stromal cells, mesenchymal stem cells, mutational analysis

## Abstract

Dystrophic epidermolysis bullosa (DEB) is an inheritable blistering disease caused by mutations in *COL7A1*, which encodes type VII collagen. To address the issue of genotype-phenotype correlations in DEB, analyzing the consequences of *COL7A1* mutations using mRNA is indispensable. Herein we established a novel method for testing the effect of mutations in DEB using *COL7A1* mRNA extracted from peripheral blood mononuclear cells (PBMCs). We investigated the consequences of four *COL7A1* mutations (c.6573 + 1G > C, c.6216 + 5G > T, c.7270C > T and c.2527C > T) in three Japanese individuals with recessive DEB. The novel method detected the consequences of two recurrent *COL7A1* mutations (c.6573 + 1G > C, c.6216 + 5G > T) and a novel *COL7A1* mutation (c.7270C > T) accurately. In addition, it detected aberrant splicing resulting from a *COL7A1* mutation (c.2527C > T) which was previously reported as a nonsense mutation. Furthermore, we revealed that type VII collagen-expressing cells in PBMCs have similar cell surface markers as mesenchymal stem cells; they were CD105^+^, CD29^+^, CD45^−^, and CD34^−^, suggesting that a small number of mesenchymal stem cells or mesenchymal stromal cells are circulating in the peripheral blood, which enables us to detect *COL7A1* mRNA in PBMCs. Taken together, our novel method for analyzing mutation consequences using mRNA obtained from PBMCs in DEB will significantly contribute to genetic diagnoses and novel therapies for DEB.

## 1. Introduction

Dystrophic epidermolysis bullosa (DEB) is an intractable genetic blistering skin disease caused by mutations in *COL7A1*, which encodes type VII collagen. Type VII collagen is the main component of anchoring fibrils. DEB can be inherited as either an autosomal dominant (DDEB) or recessive (RDEB) trait. Due to considerable phenotypic variability, DDEB is currently subdivided into four clinical subtypes and RDEB is classified into six clinical subtypes [1]. More than 900 pathogenic mutations distributed over the entire *COL7A1* gene have been reported. However, genotype–phenotype correlations remain unclear. Since mutations resulting in aberrant splicing account for at least 15% of all reported COL7A1 mutations [2,3,4,5,6], the consequences of these mutations should be examined in detail to address the issue of genotype-phenotype correlation to further understand the pathogenesis of DEB. 

In general, genomic DNA obtained from peripheral blood mononuclear cells (PBMCs) is used for mutational analysis of *COL7A1*. On the other hand, skin biopsy, an invasive technique, is thought to be essential to understanding the consequences of aberrant splicing at the mRNA level. Interestingly, Jiang et al. demonstrated that reverse transcription polymerase chain reaction (RT-PCR) using mRNA extracted from PBMCs successfully detects splice site mutations and elucidates the genetic effect of aberrant splicing in patients with DEB [7,8]; however, no other studies have replicated RT-PCR with PBMCs for *COL7A1* mutational analysis. In the present study, we analyzed *COL7A1* mutations using mRNA, not only genomic DNA, obtained from PBMCs of patients with RDEB and demonstrated its feasibility and accuracy in analyzing genetic consequences.

## 2. Results

### 2.1. Mutational Analysis of COL7A1 Using mRNA Obtained from PBMCs

After obtaining written informed consent, we performed *COL7A1* mutational analysis for two patients with intermediate RDEB (Patients 1 and 2) and one patient with severe RDEB (Patient 3) (Table 1). To evaluate the feasibility of using mRNA obtained from PBMCs for *COL7A1* mutational analysis, we analyzed two recurrent mutations in Japanese individuals. The consequences of these mutations were previously confirmed using mRNA obtained from keratinocytes or fibroblasts from the patient’s skin [9,10,11,12]. 

In Patient 1, compound heterozygous mutations (c.6573 + 1G > C and c.6216 + 5G > T) were detected with direct sequencing of genomic DNA. The genetic effect of the splice site mutation c.6573 + 1G > C was first described by Tamai et al. with RT-PCR using mRNA extracted from the patient’s cultured keratinocytes. They found that this mutation caused an activated cryptic splice site in intron 81 located 25 bp downstream from the normal donor splice site, which resulted in a frameshift and the generation of 106 additional amino acids before the creation of a premature termination codon (PTC) [9]. The second mutation, c.6216 + 5G > T, was first described by Kon et al. [10]. The outcome of this mutation was analyzed with RT-PCR using total RNA obtained from the blister roof by Hamada et al.; this mutation generated an aberrant mutant transcript with the inclusion of the entire intron 74 as a new exon, leading to PTC at 473 bp downstream, in exon 82 [11]. Further investigation with RNA extracted from the patient’s skin revealed that this splice site mutation yields several aberrant transcripts [12]. In the present study, RT-PCR using mRNA obtained from PBMCs detected two bands for each mutation, one normal-sized band and one larger band. The larger bands showed insertion of 25 or 80 base pairs of nucleotides corresponding to a part of intron 81 or the entire intron 74, respectively (Figure 1a). These results indicate that our novel non-invasive method of mutational analysis using mRNA obtained from PBMCs can accurately detect the consequences of splice site mutations, which are consistent with previous studies using mRNA obtained from keratinocytes or fibroblasts [9,10,11,12].

To further confirm the feasibility of this method, we next analyzed the consequence of an unreported splice site mutation of *COL7A1* in Patient 2. Direct sequencing of genomic DNA revealed two compound heterozygous mutations: c.409C > T and c.7270C > T (Figure 1b). For c.409C > T, direct sequencing of cDNA showed it was the same as the genomic DNA. On the other hand, the novel mutation c.7270C > T, which seems to cause aberrant splicing due to an arginine-to-tryptophan substitution at the last amino acid of exon 94 (p.R2424W) mutational analysis of genomic DNA, resulted in the deletion of the last 26 amino acids of exon 94 (r.7248_7273del26). As a result, there was a PTC at 57 amino acids downstream of the substitution (p.Q2417Sfs*57) according to RT-PCR analysis of the cDNA. Thus, mutational analysis using mRNA obtained from PMBCs successfully detected the genetic effect of a novel splice site mutation in *COL7A1*.

Our novel method of RT-PCR using mRNA obtained from PBMCs also revealed unexpected consequences of a *COL7A1* mutation that seemed to be a nonsense mutation. In Patient 3, conventional mutational analysis using genomic DNA revealed two compound heterozygous mutations in separate alleles (c.8053C > T and c.2527C > T) (Figure 1c). Regarding the former mutation, direct cDNA sequencing showed it was the same as the genomic DNA. The latter mutation, which was first reported by Kern and colleagues as a nonsense mutation [13], was expected to result in a substitution of an arginine residue to the stop codon at amino acid 843 (p.R843*). However, RT-PCR analysis of the cDNA showed a 62-amino acid deletion (r.2526_2587del62), which led to an arginine-to-alanine substitution at position 843 and yielded a PTC downstream of the substitution (p.R843Afs*1). Thus, mutational analysis of cDNA revealed that the c.2527C > T mutation, which was thought to be a nonsense mutation (p.R843*), resulted in out-of-frame deletion in this patient.

### 2.2. Type VII Collagen Expression in Isolated Cells Obtained from PBMCs Using the Human Mesenchymal Stem Cell Enrichment Cocktail

Next, we examined the expression level of *COL7A1* mRNA in NHEKs, NHDFs, PBMCs, or cultured human MSCs. RT-qPCR results showed that PBMCs and cultured human MSCs also express *COL7A1* mRNA, although much less than the expression levels of NHEKs and NHDFs. (Figure 2a). These findings suggest that mRNA extracted from PBMCs is available for *COL7A1* mutational analysis and that PBMCs contain some kinds of cells which express type VII collagen.

Although PBMCs include many types of cells, such as lymphocytes and monocytes, PBMCs expressing type VII collagen have not been identified. Several reports have shown that bone marrow-derived MSCs recruited to the dermal-epidermal junction have the potential to produce type VII collagen, which can ameliorate skin fragility in animal models and patients with RDEB [14,15,16]. Hence, we hypothesized that MSCs or mesenchymal stromal cells in PBMCs express type VII collagen.

To assess this possibility, we isolated MSCs from PBMCs using the RosetteSep Human Mesenchymal Stem Cell Enrichment Cocktail (Stem Cell Technologies) and cultured them with MesenPRO RS Medium (Thermo Fisher Scientific). After 7 days of culture, spindle-shaped cells were isolated and proliferated on cell culture dishes (Figure 2b). To characterize these cells isolated from PBMCs, we first examined type VII collagen expression. RT-qPCR revealed that the isolated cells expressed type VII collagen in response to transforming growth factor-*β* (TGF-*β*) stimulation (Figure 2c).

### 2.3. Cells Isolated from PBMCs Using the Human Mesenchymal Stem Cell Enrichment Cocktail Have a Similar Expression Pattern of Cell Surface Markers as Human MSCs

For further characterization of the cells isolated from PBMCs, we analyzed the expression of positive selection markers for MSCs (CD105 and CD29) and negative selection markers for MSCs (CD45 and CD34) with RT-PCR and immunofluorescence (IF). Cells isolated from PBMCs had little CD45 and abundant CD105 mRNA expression based on RT-PCR analysis (Figure 3a). The expression profile of cell surface markers was more similar to the profile of MSCs rather than PBMCs overall. 

In addition, IF showed that the isolated cells were CD105^+^, CD29^+^, CD45^−^, and CD34^−^ and expressed type VII collagen (Figure 3b). These immunological characteristics based on IF were similar to those of cultured human MSCs (Appendix A) and consistent with previously reported features of MSCs [17]. These results suggested that circulating PBMCs contain a small number of MSCs or mesenchymal stromal cells that express type VII collagen.

## 3. Discussion

This is the first report to show the expression of *COL7A1* mRNA in PBMCs and the feasibility of using PBMCs for mutational analysis and testing mutation consequences in DEB. An invasive skin biopsy has been thought to be essential to analyzing the consequences of aberrant splicing because *COL7A1* mRNA expression in PBMCs has not been evaluated in detail. In the present study, we established an accurate and non-invasive method of analyzing the genetic effects of mutations in RDEB using *COL7A1* mRNA extracted from PBMCs. First, we showed its feasibility and accuracy for analyzing the genetic effect of *COL7A1* mutations in patients with DEB. In Patient 1, our novel non-invasive method detected the outcomes of splice site mutations accurately, which is consistent with a previous study using mRNA obtained from keratinocytes or fibroblasts in skin samples [9,10,11,12]. We also demonstrated that this new method could detect the consequences of novel splice site mutations correctly. In Patient 2, a substitution at the last amino acid of exon 94 (p.R2424W) resulted in the deletion of the last 26 amino acids of exon 94 (r.7248_7273del26) and yielded a PTC (p.Q2417Sfs*57) in RT-PCR analysis of the cDNA. Although premature termination codon on both COL7A1 alleles typically results in severe RDEB, this patient presents a milder phenotype of intermediate RDEB. In addition, we corrected the effect of a *COL7A1* mutation, which were previously reported as another type of mutation. In Patient 3, mutational analysis using cDNA revealed that the c.2527C > T mutation, which was thought to be a nonsense mutation (p.R843*), resulted in an out-of-frame deletion (r.2526_2587del62) and yielded a PTC downstream of the substitution (p.R843Afs*1) as a result. Taken together, RT-PCR and direct sequencing of cDNA using mRNA obtained from PBMCs are useful and non-invasive methods that can accurately confirm the consequences of pathogenic mutations in patients with DEB. 

The present study also revealed that cells isolated from PBMCs using the Human Mesenchymal Stem Cell Enrichment Cocktail express type VII collagen and positive selection markers for MSCs (CD45 and CD34), suggesting that small numbers of MSCs or mesenchymal stromal cells that express type VII collagen are circulating in the peripheral blood. Although the actual function of type VII collagen-expressing MSCs and/or mesenchymal stromal cells remain unclear, it has been reported that locally injected or intravenously administered MSCs are associated with clinical improvement in patients with RDEB, with the transient restoration of type VII collagen at the basal membrane zone [16,18,19,20,21]. These findings also support our results that MSCs can express type VII collagen. The existence of type VII collagen in the peripheral blood will be helpful for developing novel diagnostic and therapeutic methods for DEB. Although we isolated a small number of *COL7A1* mRNA-expressing cells from PBMCs in healthy individuals, Tamai and colleagues reported that injured or necrotic epidermal cells in RDEB release high levels of high mobility group box-1 (HMGB-1), which recruits bone marrow-derived circulating mesenchymal cells to migrate and supply functional type VII collagen to regenerate the skin in RDEB [22]. Therefore, more *COL7A1* mRNA-expressing mesenchymal cells might circulate in the peripheral blood in patients with RDEB, which enables us to perform *COL7A1* mutational analysis using mRNA obtained from PBMCs of patients with RDEB. 

Fibrocytes are a distinct population of bloodborne fibroblast-like cells. They have been reported to comprise 0.1–0.5% of PBMCs. They circulate in the peripheral blood and rapidly enter sites of tissue injury to play a pivotal role in wound healing processes. Fibrocytes have an adherent, spindle-shaped morphology. They express some products of fibroblasts, such as type I and III collagen, *α*-smooth muscle actin, and fibronectin in response to TGF-*β* stimulation. Although the ability of fibrocytes to produce and secret type VII collagen remains unclear, they could represent a population of type VII collagen-expressing cells in PBMCs based on their morphology, protein expression profile, and responsiveness to TGF-*β*, as mentioned above. However, fibrocytes display a unique cell surface phenotype of collagen I^+^/CD11b^+^/CD13^+^/CD34^+^/CD45RO^+^/MHC class II^+^/CD86^+^ [23], whereas IF and RT-PCR evaluations in the present study demonstrated that cultured cells obtained from PBMCs expressing type VII collagen do not express CD45 and CD34. Therefore, fibrocytes would not be the main population of PBMCs expressing type VII collagen. 

In conclusion, the present study showed the expression of *COL7A1* mRNA in PBMCs and the feasibility of using PBMCs to analyze the consequences of *COL7A1* mutations in patients with DEB. We also demonstrated that circulating MSCs or mesenchymal stromal cells could be the main components of type VII collagen-expressing cells in PBMCs. Although the actual function of those cell types remains to be elucidated, the expression of type VII collagen in the peripheral blood is helpful for developing novel diagnostic and therapeutic methods for DEB.

## 4. Materials and Methods

### 4.1. Mutational Analysis

Two patients with intermediate RDEB and one patient with severe RDEB participated in this study (Table 1). This study was approved by the ethics committee of the Hirosaki University Graduate School of Medicine (2020—146—2). Blood samples were collected after obtaining written informed consent. Genomic DNA was extracted using standard protocols with QIAamp DNA Blood Maxi Kit (QIAGEN, Hilden, Germany). Polymerase chain reaction (PCR) fragments were amplified with previously described primers spanning all 118 exons and exon-intron boundaries of *COL7A1* [24]. Mutations were detected with a direct sequence using an ABI PRISM 3130 Genetic Analyzer (Thermo Fisher Scientific, Waltham, MA, USA). Messenger RNA (mRNA) was also isolated from PBMCs using NucleoSpin RNA Blood (Takara Bio, Shiga, Japan) and reverse transcribed using the PrimeScript™ II 1st strand cDNA Synthesis Kit (Takara Bio) to obtain complementary DNA (cDNA). The cDNA was amplified with RT-PCR for direct sequencing. The restriction fragment length polymorphism (RFLP) method was used to detect mutations and its consequences.

### 4.2. Cell Culture

Normal human epidermal keratinocytes (NHEKs) were cultured in a keratinocyte serum free medium (Thermo Fisher Scientific) with 50 µg/mL of bovine pituitary extract (Thermo Fisher Scientific) and 5 ng/mL of recombinant epidermal growth factor (Thermo Fisher Scientific). Normal human dermal fibroblasts (NHDFs) were cultivated in Dulbecco’s Modified Eagle Medium (DMEM) (Thermo Fisher Scientific) with 10% fetal bovine serum (FBS) (Thermo Fisher Scientific), 1% Antibiotic-Antimycotic (Thermo Fisher Scientific), 4 mM of L-glutamine (Thermo Fisher Scientific), and 1 mM of sodium pyruvate (Thermo Fisher Scientific). Human mesenchymal stem cells (MSCs) (PromoCell, Heidelberg, Germany) were cultured in an MSC growth medium (PromoCell). Cells were cultivated in a humidified incubator at 37°C and 5% CO_2_. NHDFs were starved in media without FBS for 24 h and then treated with 1 nM of recombinant human transforming growth factor-*β*1 (Active TGF-*β*; Cell Signaling Technology, Danvers, MA, USA) for 48 h.

### 4.3. Isolation of Mesenchymal Stromal Cells from PBMCs

Mesenchymal stromal cells were isolated from PBMCs of healthy donors according to a previously reported method [25]. Briefly, 15 mL of peripheral blood and an equal volume of phosphate buffered saline (PBS) containing 2% FBS (2% PBS) were mixed and slowly layered on top of LSM^TM^ Lymphocyte Separation Medium (MP Biomedicals, Irvine, CA, USA). After 20 min of centrifugation at 800× *g*, the layers of plasma and the lymphocyte separation medium were removed. The mononuclear cell layer was collected into another tube, mixed with 2% PBS in 1:1 dilution, and centrifuged at 200× *g* for 10 min. The supernatant was discarded. The enriched cell pellet (concentrated mononuclear cells) was resuspended with 4 mL of additional peripheral blood from the same patient as red blood cells are necessary to apply 200 μL of the RosetteSep Mesenchymal Stem Cell Enrichment Cocktail (Stem Cell Technologies, Vancouver, BC, Canada); this antibody cocktail mix contains bi-specific antibody complexes against red blood cells (glycophorin A) and cells positive for CD3, CD14, CD19, CD38, or CD66b. These antibody complexes crosslink the unwanted cells with red blood cells in the sample, causing them to pellet together when centrifuged. After 20 min of incubation at room temperature, 8 mL of 2% PBS containing 1 mM of ethylenediaminetetraacetic acid (EDTA) (2% PBS-EDTA) was added. The mixture was layered on top of LSM^TM^ Lymphocyte Separation Medium. After 25 min of centrifugation at 300× *g*, the enriched cells were washed in 2% PBS-EDTA twice and then cultured at a concentration of 3.0 × 10^6^ cells in 12-well plates with MSC growth medium (PromoCell).

### 4.4. Reverse Transcription Quantitative Polymerase Chain Reaction (RT-qPCR)

RNA was extracted from NHEKs, NHDFs, cultured MSCs, or PBMCs using the RNeasy Plus Mini Kit (QIAGEN) according to the manufacturer’s instructions. RNA was reverse-transcribed to cDNA using the First Strand cDNA Synthesis Kit (Thermo Fisher Scientific). qPCR analysis was performed with agarose gel electrophoresis or SYBR Green labeling and a CFX96 Real-Time system (Bio-Rad, Hercules, CA, USA). The primers used were as follows: *CD105*F: CAACAATGCAGATCTGGACCAC; *CD105*R: CTTTAGTACCAGGGTCATGGC; *CD45*F: GCAGTTCATGCAGCTAGCAAGT; *CD45*R: AAGGTGTTGGGCTTTGCCCTGT; *COL7A1*F: CACGGGGACCTCCGGAGCTAT; *COL7A1*R: ATCTCGTCCTCGGGGGCCAACA; *GAPDH*F: CCCATCACCATCTTCCAG; *GAPDH*R: ATCACCTTGCCCACAGCG.

### 4.5. Immunofluorescence of Peripheral Blood-Derived Mesenchymal Stromal Cells

Adherent mononuclear cells obtained from PBMCs using MSC enrichment cocktails were cultured on coverslips. After washing with PBS, the cells were incubated with paraformaldehyde for 15 min and with 0.5% of Triton X-100 for 5 min for fixation. The cells were washed with PBS three times and then blocked with 1% bovine serum albumin for 30 min. The cells were incubated with primary antibodies listed in Table 2 for 1 h, followed by incubation with fluorescein isothiocyanate (FITC) or Alexa 647 conjugated secondary antibodies in the dark. After mounting with 4′,6-diamidino-2-phenylindole (DAPI), cells were analyzed using a fluorescence microscope.

## Figures and Tables

**Figure 1 ijms-22-13369-f001:**
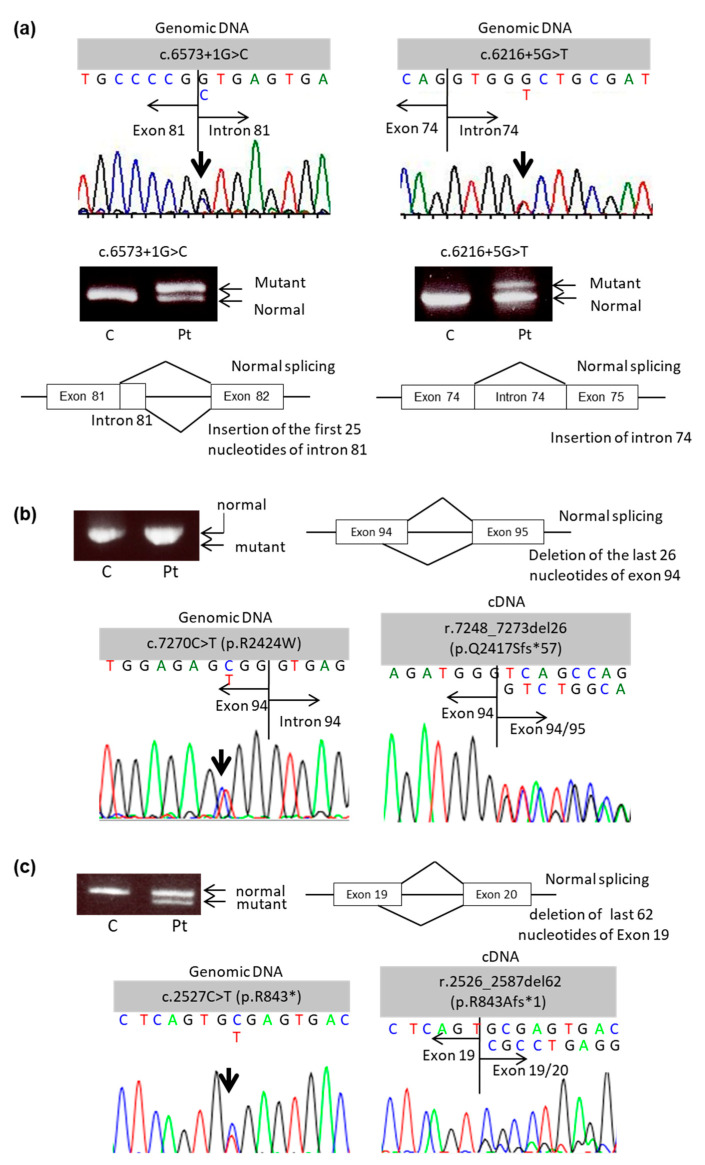
*COL7A1* mutational analysis results. Genomic DNA and messenger RNA (mRNA) were obtained from patient peripheral blood mononuclear cells (PBMCs). Results of agarose gel electrophoresis for the products of reverse transcription polymerase chain reaction (RT-PCR) and results of direct sequencing analysis of genomic DNA and complementary DNA (cDNA) are presented. (**a**) Patient 1, (**b**) Patient 2, and (**c**) Patient 3.

**Figure 2 ijms-22-13369-f002:**
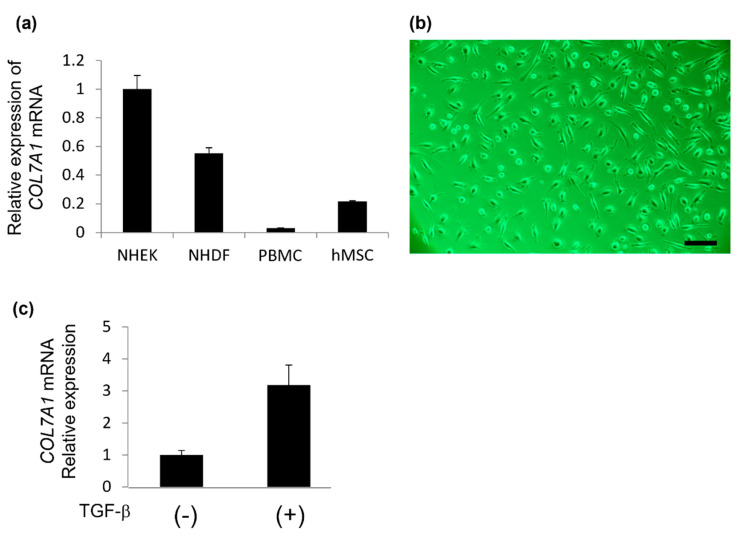
Type VII collagen expression in cells isolated from PBMCs obtained using the Human Mesenchymal Stem Cell Enrichment Cocktail. (**a**) *COL7A1* mRNA expression of cultured normal human epidermal keratinocyte (NHEK), normal human dermal fibroblast (NHDF), Peripheral blood mononuclear cell (PBMC), and human mesenchymal stem cell (hMSC) was analyzed using reverse transcription quantitative polymerase chain reaction (RT-qPCR). (**b**) The isolated cells were cultured in mesenchymal stem cell culture medium for 14 days. Scale bar, 100 μm. (**c**) *COL7A1* mRNA expression in isolated cells. Cells were treated with or without 1 nM of transforming growth factor- (TGF-β). Quantitative analysis of *COL7A1* mRNA expression was performed using RT-qPCR.

**Figure 3 ijms-22-13369-f003:**
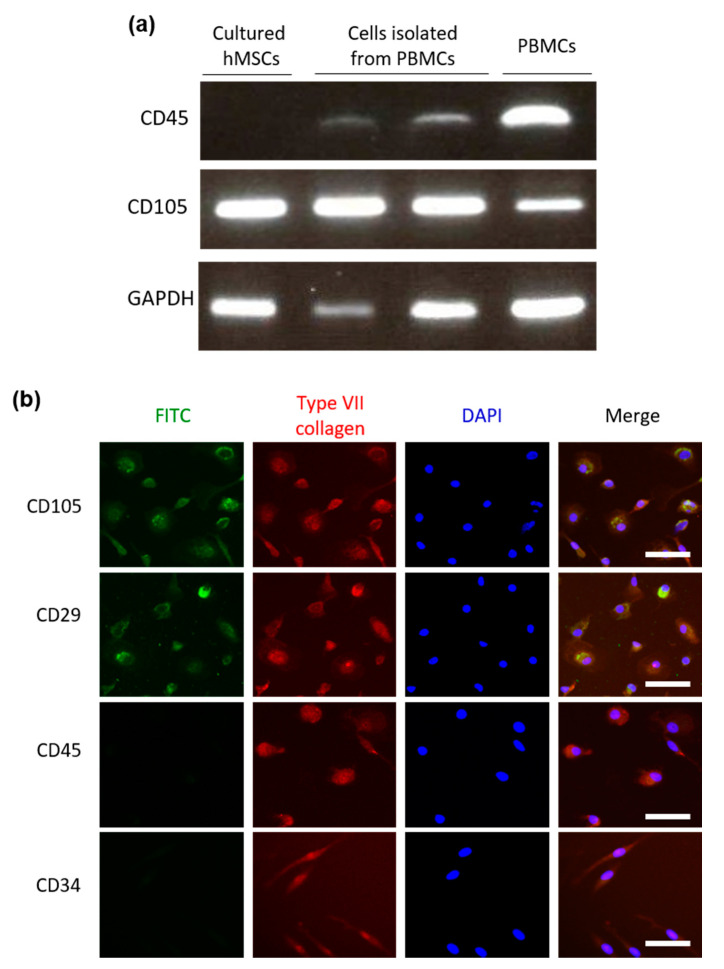
Immunological characteristics of cells isolated from peripheral blood mononuclear cells (PBMCs) using the Human Mesenchymal Stem Cell Enrichment Cocktail. (**a**) Expression of lineage markers CD105 or CD45 was investigated using semi-quantitative reverse transcription polymerase chain reaction (RT-PCR). Left lane, cultured human mesenchymal stem cells (hMSCs); middle lanes, cells isolated from PBMCs; right lane, PBMCs. (**b**) Immunophenotyping of cultured and isolated cells. Immunofluorescence staining for lineage markers CD105, CD29, CD45, and CD34 (green); type VII collagen (red); and nuclear component (blue). DAPI, 4′,6-diamidino-2-phenylindole; FITC, fluorescein isothiocyanate. Scale bar, 100 μm.

**Table 1 ijms-22-13369-t001:** Phenotypes and genotypes of patients with recessive dystrophic epidermolysis bullosa.

Patient No.	Age (years)	Sex	Clinical Subtype of DEB	Genotype (Genomic DNA)	Expected Consequence	Genotype (cDNA)	Expected Consequence
1	27	F	Intermediate RDEB	c.6573+1G > C	Splice site mutation		Insertion of 25 nucle- otides of intron 81
				c.6216+5G > T	Splice site mutation		Insertion of intron 74
2	40	M	Intermediate RDEB	c.409C > T (p.R137*)	Nonsense mutation	c.409C > T (p.R137*)	Nonsense mutation
				c.7270C > T (p.R2424W)	Splice site mutation	r.7248_7273del26 (p.Q2417Sfs*57)	Splice site mutation leading to PTC
3	19	M	Severe RDEB	c.8053C > T (p.R2685*)	Nonsense mutation	c.8053C > T (p.R2685*)	Nonsense mutation
				c.2527C > T (p.R843*)	Nonsense mutation	r.2526_2587del62 (p.R843Afs*1)	Splice site mutation leading to PTC

M, male; F, female; RDEB, recessive dystrophic epidermolysis bullosa; PTC, premature termination codon.

**Table 2 ijms-22-13369-t002:** Antibodies used in immunofluorescence.

Antibody	Host Animal	Dilution	Manufacturer (Location)
Anti-collagen type VII (LH7.2)	Mouse	1:50	Sigma-Aldrich (St. Louis, MO, USA)
Anti-CD105	Rabbit	1:200	Abcam (Cambridge, UK)
Anti-CD29	Rabbit	1:50	Abcam (Cambridge, UK)
Anti-CD45	Rabbit	1:200	Abcam (Cambridge, UK)
Anti-CD34	Rabbit	1:50	Abcam (Cambridge, UK)
Anti-mouse IgG - Alexa Fluor 647	Donkey	1:200	Abcam (Cambridge, UK)
Anti-rabbit IgG - FITC	Goat	1:200	Sigma-Aldrich (St. Louis, MO, USA)

HRP, horseradish peroxidase; IgG, immunoglobulin G.

## Data Availability

The datasets used and/or analyzed during the current study are available from the corresponding author on reasonable request.

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
