# Peer review of "Availability of mRNA Obtained from Peripheral Blood Mononuclear Cells for Testing Mutation Consequences in Dystrophic Epidermolysis Bullosa"

_ijms, 2021, doi:10.3390/ijms222413369_

Round 1
Reviewer 1 Report
In this paper the Authors report on the efficacy of investigating the effects of COL7A1 mutations on mRNA splicing using directly total RNA purified from patient PBMC rather than skin cultured cells or biopsy.
This is a possibility well known to scientists working on genotype-phenotype correlation in dystrophic epidermolysis bullosa (DEB). However, the study has elements of originality in the part that concern the typing of circulating blood cells that most contribute to the synthesis of COL7A1 RNA.
Major concerns:
- I suggest a change in the Title. It is slightly misleading because the mutational analysis referred to is not a mutational screening method but simply an alternative way of testing the consequences of certain mutations previously identified with the gold standard of massive DNA sequencing on panels. I would suggest saying "... for testing mutation consequences in DEB" -The percentage of COL7A1 splice site mutations is certainly higher than that indicated (17%) in the Introduction. The quoted paper is quite dated (Varki 2007). Percentage around 20-22% is reported in more recent papers.
- I was wondering whether discovering the true mutational effect of c.7270C>T in patient 2 has helped to refine better the phenotype-genotype correlation. In theory, the p.Q2417Afs*57 effect is more severe than missense p.R2424W. Mutation combination in this patient (truncating mutations in both alleles) is expected to result in severe RDEB rather than intermediate RDEB.
- The grammar and syntax of the English language need improvement. I suggest a review by a native speaker. For example, the sentence on page 5 lines 126-128 must be set more correctly.
Reviewer 2 Report
The paper describes the potential use of mRNA from peripheral blood mononuclear cells to conduct mutational analysis in dystrophic epidermolisis bullosa. The authors have designed a scientifically sound study, with logically planned stages. The results are clearly presented in a logical succession; it is noteworthy that the authors discussed their results versus other previous studies thus enlarging the readers' understanding of the topic. The formulated conclusions are supported by the experimental results. The paper is written in a fluent language and is easily readable yet at academic standards. I fully recommend the publication of the paper in its current form.
